# Emotional Attitude to Own Appearance and Appearance of the Spouse: Analysis of Relationships with the Relationship of Spouses to Themselves, Others, and the World

**DOI:** 10.3390/bs10020044

**Published:** 2020-01-29

**Authors:** Tatyana A. Vorontsova, Vera A. Labunskaya

**Affiliations:** Department of Social Psychology, Academy of Psychology and Education, Southern Federal University, 13, M. Nagibina Street, 344038 Rostov-on-Don, Russia; vlab@aaanet.ru

**Keywords:** appearance, attitudes toward appearance, spouse, registered marriage, unregistered marriage

## Abstract

The study analyzed the relationship of attitudes toward one’s appearance and appearance of the partner with attitude toward the own personality and that of the other persons’ in married men and women. The empirical object of the study included 52 married couples in a registered (26 couples) and unregistered (26 couples) marriage with a duration from 5 months to 26 years (M = 7.31; SD = 6.78). The age of the respondents was 20–45 years old (M = 30.26; SD = 7.31; all are residents of the Russian Federation; Russians). Methods included the following questionnaires: (1) “History of the couples’ relationships”; (2) “Estimated and informative interpretation of one’s appearance and its compliance with gender–age constructs”; (3) “Color test of relationships”; (4) “Method of diagnosing interpersonal relationships”; (5) “Fundamental Interpersonal Relations Orientation-Behavior questionnaire”. Empirical data were analyzed with Spearman correlation analysis, Mann–Whitney U-test, Kruskal–Wallis H-test. The results were as follows: (1) in men and women the attitude to their appearance is related to the attitude to themselves; attitude to the spouse’s appearance is associated with the attitude to him or her; (2) in women, the assessment of their appearance is related to the attitude to their appearance; in men, the assessment of their appearance is related to the attitude to appearance of their spouses; (3) women’s attitude to their appearance is associated with the need for inclusion, while in the men’s case it is associated with the need for love; (4) men who emotionally reject their mothers are dissatisfied with the appearance of their wives.

## 1. Introduction

The study is performed within the framework of the socio-psychological approach to the person’s appearance by Labunskaya [1], in which the appearance (AP) of a person is defined as “a constructed form of objectivization of a person’s inner world, as a phenomenon reflecting various stages of the life journey based on the dynamic, variable interrelationships of three components: (1) physical component, (2) social appearance, (3) expressive behavior” [1] (p. 202). Within this approach, the task is to identify a set of factors influencing the attitude toward AP in various interaction contexts. As one specific interaction context, this article discusses family and marriage relationships.

In most works devoted to the study of AP of spouses, its role in choosing the marriage partner is noted [2,3]. A study by Gui [2], conducted in large cities in China, showed that younger age and attractive AP are considered as assets of women, and those of men are education and higher incomes. In the study by Russian scientists Karapetyan and Zhiltsova [4], it is also showed that women, more than men, are preoccupied with their external attractiveness to spouses.

The role of AP in the satisfaction with marriage [5] is also being studied: the highest appreciation of own visual attractiveness corresponds to the highest level of satisfaction with marriage. The revealed pattern is typical both for men and women. In a number of works, a different significance of an attractive AP is noted depending on the length of time the couple lives together. The study by Evtukh [6] showed that for couples being married for a shorter time, the significance of the external attractiveness of the spouse is higher than it is for partners in longer relationships. In the work by Ivanchenko and Shchepina [7], it was found that the attractiveness of one’s own AP and that of the partner during the first years of life together is more important for women than for men. At the same time, in the work by Ma-Kellams et al. [8], it is demonstrated that the men being assessed as the most attractive ones, divorce their spouses more often than others, with a short duration of marriage. Moreover, in the work by McNulty et al. [9], the relative difference between the levels of attractiveness of partners turned out to be most important for predicting family behavior: thus, both spouses behaved more positively in relationships where wives were more attractive than their husbands, and more negatively in relationships where husbands were more attractive than their wives.

In a number of works, it is shown that the attitude toward the partner’s AP is a prognostic “marker” of various disturbances in the family functioning. For example, Alirezaei et al. [10] showed that the satisfaction/dissatisfaction with the spouse’s appearance is one of the factors affecting sexual dysfunction in infertile women. Pournaghash-Tehrani [11] found that the attitude toward the wife’s AP influences predicting the men’s response to violence from the part of the spouse.

Furthermore, studies have shown the connection between the self-assessment of AP by husbands and wives, as well as between the self-assessment of AP by husbands/wives themselves and their AP assessment by the other spouse [12]. This pattern was detected for elderly couples (over 60 years old).

## 2. The Present Study

### 2.1. The Goals and Objectives of the Present Study

The conducted analysis provides an opportunity to state that there is a lack of data on the interconnection of attitude to own AP and AP of the spouse with other elements of the system of relations: to the own personality, other persons, the world, as well as social needs. The analysis also showed that the attitude toward the own AP and the AP of partner in the context of legitimizing relationships in a couple (registered/unregistered marriage) is almost unstudied. At that, the current state of the institution of marriage is that the share of unregistered marriages increases, and the popularity of the traditional registered marriage falls [13,14]. An unregistered marriage is an informal one (not registered by civil law). In Russia, according to various estimates, the share of such marriages is up to 60% (for couples aged 18–24 years). This trend is observed in Europe and the United States [13,14]. We assume that the emotional attitude of the spouses to their appearance and the appearance of the partner will participate in the decision of the spouses to legitimize the marriage. The research design is developed based on the concept of relations by Myasishchev [15], in which relationships are considered as an integral system of an individual’s relations with different sides of objective reality, which includes three interrelated components: man’s attitude toward the own personality, other persons, and the world. The study analyzed the relationship of emotional, partially unconscious attitudes toward one’s AP and AP of the partner with basic social and psychological needs (in participation, control, love), attitude toward the own personality, other persons (partner, mother/father, children), to important phenomena and concepts (family, marriage, love, freedom, money, divorce), and with types of interpersonal relationships in married men and women. Moreover, there was the task of identification of the influence of the type of marriage (registered/unregistered) on the attitude toward own AP and the partner’s AP. We also decided to check (on our sample) the pattern found in the work by Oh and Damhorst [12] for older couples that the relationship of the spouses may be interrelated to own AP and AP of the partners.

### 2.2. Hypotheses

As the study hypotheses, the following assumptions were used: (1) The attitude toward own AP and AP of the partner may be differently determined by the components of the personality’s relations system and the expressiveness of the socio-psychological needs of married men and women. (2) Emotional attitudes toward own AP and AP of the partner at spouses in a registered and an unregistered marriage may differ. (3) The relationship of the spouses (age up to 45 years) may be interrelated to own AP and AP of the partners.

### 2.3. Procedure

The empirical object of the study included 104 people (52 women and 52 men)—52 married couples in a registered (26 couples) and unregistered (26 couples) marriage with a duration from 5 months to 26 years (M = 7.31; SD = 6.78). The age of respondents was 20–45 years old (M = 30.26; SD = 7.31; they were residents of the Russian Federation from the big city, Rostov-on-Don). Data collection was carried out by filling out paper forms of questionnaires by the spouses. Each of the spouses filled out their own copy of the tests separately from the partner.

All subjects gave their informed consent for inclusion before they participated in the study. The study was conducted in accordance with the Declaration of Helsinki, and the protocol was approved by the Ethics Committee of the Academy of Psychology and Education of the Southern Federal University (Protocol 1 of 30 January, 2019).

### 2.4. Methods

The following techniques were used in the work:

(1) The questionnaire “History of the couples’ relationships” (Shkurko and Lomova) [16] was used for recording the socio-demographic features of the study participants (age, duration of relationships, presence/absence of official registration, etc.). (2) The method “Estimated and informative interpretation of one’s AP and its compliance with gender–age constructs” by Labunskaya [1] was used for studying the spouses’ self-assessment of the various components of their AP (face, bodily structure, expressive behavior), integral self-assessment (attractiveness, AP sexuality, AP integral assessment), as well as the parameter of satisfaction with own AP. (3) Color test of relationships by Etkind (short version) [17]. This test was used for diagnosing the emotional, partially unconscious attitude (reaction of acceptance/rejection, recognition/non-recognition) of spouses toward themselves, other people (husband/wife, mother/father, children), their AP and AP of the partner, important phenomena and concepts (our couple, family, unregistered marriage, registered marriage, commitment, responsibility, love, freedom, money, divorce). (4) The method of diagnosing interpersonal relationships by Leary in the adaptation of Sobchik [18] was used for the identification of the following types of attitudes to other people: (1) dominating and leading; (2) independently dominant; (3) straight and aggressive; (4) incredulous and skeptical; (5) submissively shy; (6) dependent and obedient; (7) collaborative and conventional; and (8) responsibly generous. (5) “Fundamental Interpersonal Relations Orientation-Behavior questionnaire” by Schutz [19] adapted by Rukavishnikov [20] was used to study the expressiveness of the three basic social needs (participation, control, love) at the level of expressed behavior and the one required from others.

### 2.5. Statistics

For data processing, the following was used: Spearman correlation analysis, Mann–Whitney U-test, Kruskal–Wallis H-test. These methods were used for processing data obtained through questionnaires based on metric scales. The data were processed with SPSS 24. P-values ≤ 0.05 were considered statistically significant.

## 3. Results

### 3.1. Analysis of the Relationship of Attitudes toward Own AP and AP of the Partner with the Attitude of the Spouses toward Themselves, Other Persons, and the World

In order to prove the first hypothesis, the Spearman correlation analysis was individually applied to the data obtained for the subsamples of men and women (Table 1 and Table 2).

The analysis of the data given in Table 1 and Table 2 allows a number of conclusions:

(1) There were similar interrelations in women and men: an emotional, partially unconscious attitude toward own AP was closely connected with an emotional attitude toward the own personality (women r = 0.438 **, *p* = 0.001; men r = 0.310 *, *p* = 0.025); the emotional attitude toward the partner’s AP was closely connected with the attitude toward the partner (women r = 0.576 **, *p* = 0.000; men r = 0.668 **, *p* = 0.000);

(2) The opposite interrelationships of the personal assessment of their AP and unconscious attitude toward own AP and AP of the spouse were found. Thus, in women, assessment of own AP was directly related to the emotional attitude toward own AP (Assessment of own bodily structure and Attitude of women to their AP, r = 0.312 *, *p* = 0.024; Assessment of own AP development and Attitude of women to their AP, r = 0.391 **, *p* = 0.004; Assessment of expressive behavior and Attitude of women to their AP, r = 0.483 **, *p* = 0.000; Degree of acceptance of own reflected AP and Attitude of women to their AP, r = 0.289 *, *p* = 0.038; Assessment of AP conformity to gender and Attitude of women to their AP, r = 0.336 *, *p* = 0.015; Integral assessment of own AP and Attitude of women to their AP, r = 0.308 */0.026). Men showed a paradoxical relationship: assessment of own AP (person, bodily structure, integral assessment) was interconnected not with the attitude toward own AP, but with the attitude toward the AP of the spouse (Assessment of own face and Attitude of men to AP of the spouse r = 0.304 *, *p* = 0.028; Assessment of own bodily structure and Attitude of men to AP of the spouse, r = 0.566 **, *p* = 0.000; Integral assessment of own AP and Attitude of men to AP of the spouse, r = 0.296 *, *p* = 0.033).

(3) Only women had a number of interrelations with their AP and the AP of the partner with the types of interpersonal relationships: the women who valued their own appearance had a dominating and leading type of interpersonal relations (Dominating and leading type of interpersonal relationships and Attitude of women to their AP, r = 0.293 *, *p* = 0.035), while women with cooperative and conventional and responsibly generous types of interpersonal relationships emotionally valued the appearance of their husbands (Collaborative and conventional type of interpersonal relationships and Attitude of women to the spouse’s AP, r = 0.379 **, *p* = 0.006; Responsible and generous type of interpersonal relationships and Attitude of women to the spouse’s AP, r = 0.279 *, *p* = 0.045).

(4) It was found that in women, the attitude toward their AP was related to the need for participation and the level of behavior required from others (Attitude of women to their AP and Need for inclusion in social groups (required behavior), r = 0.536 **, *p* = 0.000), and for men, it was related to the need for love from other people (Men’s attitude toward own AP and Need to be loved, r = 0.316 *, *p* = 0.023). The men also showed a seemingly paradoxical inverse proportion of the attitude to AP of the spouse and the expressiveness of the need to love: only the men who were very selective in establishing close relationships with other people emotionally accepted the appearance of the spouse (Attitude of men to AP of the spouse and Need to love, r = 0.301 *, *p* = 0.030).

(5) Furthermore, a number of interrelationships were found within the parameters of unconscious emotional relationships. For women, it was between the emotional attitude to their AP and the attitude to their children (r = 0.294 *, *p* = 0.034), as well as to a civil marriage (r = 0.299 *, *p* = 0.031), between the relation to AP of the partner and the attitude toward the registered marriage (r = 0.280 *, *p* = 0.045). For men, the attitude toward AP of the spouse was interrelated with the attitude toward the family (r = 0.338 *, *p* = 0.014), the category of “love” (r = 0.503 **, *p* = 0.000), and their mothers (r = 0.275*, *p* = 0.048). That is, a man who emotionally accepts his mother, as well as the categories of family and love, considered his wife attractive and emotionally accepted her AP.

### 3.2. Comparative Analysis of Emotional Attitude to Own AP and AP of the Spouse in Registered and Unregistered Marriages

In order to identify the influence of the type of marriage on the emotional attitude to own AP and AP of the spouse, Mann–Whitney test was used. The comparative analysis was conducted for data obtained from samples of men and women, each of which was divided into two subgroups according to the “marriage type” criterion: registered/unregistered marriage. A number of differences were found in the emotional attitude toward own AP and AP of the partner in spouses in a registered and unregistered marriage (Table 3).

It was found that the attitude toward own AP was less positive in women in a registered marriage, and, on the contrary, own AP was more accepted and positively evaluated by women in an unregistered marriage. At the same time, men in a registered marriage were more positive toward AP of their wives than men in an unregistered marriage.

In connection with the results obtained, there is a question: Is such an attitude toward own AP (for women) or the wife’s AP (for men) the cause or consequence of the marriage legitimization (registration)? To answer this question, we compared: (1) the attitude to own AP in women in a registered marriage; (2) the attitude to the wife’s AP in the men in the registered marriage at various stages of marriage. Each subsample of men and women in a registered marriage was divided into three groups: (1) with marriage duration from 0.5 to 10 years; (2) from 11 to 20 years; and (3) from 21 to 30 years. The comparison was made according to the Kruskal–Wallis criterion (Table 4).

The obtained data show that with an increase in the duration of the registered marriage, the emotional attitude to own appearance in women becomes less positive, and the attitude of men to AP of their wives does not change in relation to the marriage duration. This allows the following conclusion: (1) a lower level of attitude toward own appearance in women in an registered marriage, compared with women in an unregistered marriage, is a consequence of the legitimization of relations (the conclusion of a registered marriage); (2) a higher level of attitude toward the appearance of the partner in men in an registered marriage, compared with ones in an unregistered marriage, is a factor of the legitimization of relations (preceding the marriage conclusion).

Obviously, the marriage duration and the age of women participating in the study are related variables. If we conclude that the experience of marriage affects the dynamics of the attitude of women to their appearance, then it is necessary to exclude the parallel influence of age. We calculated the correlation coefficients of age and attitude to own appearance individually for samples of women in registered (r = −0.548 ** at *p* = 0.004) and unregistered marriages (r = 0.105 at *p* = 0.609). A paradoxical result was detected: women who were officially married were exposed to the age dynamics of attitude to their AP, namely: as the age increases, their emotional acceptance of own AP decreases, and women in an unregistered marriage did not show such a tendency.

### 3.3. Analysis of Interrelations of Attitudes to Own AP and AP of the Spouse

Testing the third hypothesis was conducted using the Spearman correlation analysis, first on the entire sample, then individually on the samples of spouses in a registered and unregistered marriage. Four indicators were analyzed: (1) the attitude of the wife to her AP; (2) the attitude of the husband to his AP; (3) the attitude of the wife to the husband’s AP; (4) the attitude of the husband to the wife’s AP. No significant relationships were found both in the total sample, and in the sample of spouses in an unregistered marriage. The sample of spouses in a registered marriage showed a connection between the indicators of “wife’s attitude to own AP” and “husband’s attitude to own AP” (r = 0.430, *p* = 0.028).

## 4. Discussion

The study showed a contradiction, which consists in the fact that own AP is less accepted and positively evaluated by women in a registered marriage; at the same time, men in a registered marriage relate to AP of their wives significantly more positively than men in an unregistered marriage. We have called this phenomenon “the asymmetry of attitudes toward own AP and the partner’s attitude thereto”, which is characteristic for women in a formal marriage.

The role of attitude toward own AP and AP of the spouse in the context of the couple’s relations legitimization was revealed: the decrease of acceptance of own AP by women in a registered marriage is a consequence of the relations’ legitimization (conclusion of a formal marriage); the acceptance of the woman’s AP by a man is a factor of the relations’ legitimization (preceding the marriage conclusion). A different dynamic of attitudes toward own AP associated with the type of marriage was also discovered: for women in a formal marriage, the emotional acceptance of their AP decreases with age, for women in an informal marriage, there is no such tendency.

A special role of the attitude of a man to his mother in the assessment of his wife’s AP was confirmed, which was found in the work by Bereczkei et al. [21].

The sample of spouses in a registered marriage showed the relationship of the wife’s attitude to her AP, and her husband’s relationship to his AP was found. This conclusion is a confirmation of the pattern found by Oh and Damhorst [12] on a sample of older couples (over 60 years old). At the same time, the second pattern discovered by scientists and concerning the reciprocity of self-assessment and the AP assessment was not confirmed for younger couples (under 45 years old).

The data obtained during the study confirm the results of the study by Bale and Archer [22] on the contribution of assessing own attractiveness to the overall self-assessment. In the terminology of our study, the positive attitude of the spouses toward their AP implies a positive self-attitude, and a positive attitude toward the husband’s/wife’s AP implies a positive attitude toward this spouse.

The data we obtained, which testify to the complex interrelationships of attitudes of spouses to own AP and AP of the partner with self-assessment, attitude to the partner, other meaningful persons (mothers), social needs, and types of interpersonal relations, confirm the findings of Liechty and Yarnal [23] on the influence of interpersonal relations on the formation of the body image at different stages of life, the components of which are the attitude to the appearance and self-esteem.

In general, the conducted study shows the functional significance of AP in the field of family and marriage, romantic relationships, and the complex multi-level determination of the attitude toward AP [24].

### 4.1. Limitations

The limitation of this study is, first of all, the limited sample size. Increasing the sample size would make the conclusions more valid. Secondly, the age of spouses was limited to 45 years. Therefore, the findings cannot be fully applied to the spouses who are at the next stage of life. Further research in this direction involves the analysis of the identified relationships in samples of older respondents. Third, the use of the Spearman correlation coefficient, on the one hand, can be considered as a certain limitation of the study, since it does not allow identifying the cause-and-effect relationship between the studied variables. However, on the other hand, its application corresponds to the theoretical grounds of the study set out in Section 2.1., which involve analyzing the interrelations of the elements of an individual’s relationship system, rather than their mutual influence. Fourth, the sample of the study was not completely random, since the target sample method was used when selecting respondents. In the future, a larger study using randomization procedures is expected. Fifth, all respondents live in a big Russian city (more than 1 million people), which suggests caution in using the obtained patterns to represent residents of other regions, small cities, and megalopolises.

### 4.2. Practical Implications

The obtained data can be used in systemic family psychotherapy, when analyzing and correcting the relationship of the spouses to their AP and the AP of the partner. Moreover, the attitude to the AP of a woman in a registered marriage can be a possible factor of the couples’ relationship destabilization. The last statement needs additional verification.

## Figures and Tables

**Table 1 behavsci-10-00044-t001:** Significant correlation relationships of attitudes to their appearance (AP) and the AP of women with their attitudes toward themselves, others, and the world.

Dimension	Variables	Attitude of Women to Their AP	Attitude of Women to the Spouse’s AP
Assessment of components of own AP	Assessment of own bodily structure	0.312 */0.024	0.124/0.381
	Assessment of own AP development	0.391 **/ 0.004	0.134/0.342
	Assessment of expressive behavior	0.483 **/0.000	0.228/0.104
	Degree of acceptance of own reflected AP	0.289 */0.038	0.029/0.840
	Assessment of AP conformity to gender	0.336 */0.015	0.271/0.052
	Assessment of AP compliance with gender roles	0.014/0.924	0.432 **/0.001
	Integral assessment of own AP	0.308 */0.026	0.119/0.401
Expressiveness of social needs	Need for inclusion in social groups (required behavior)	0.536 **/0.000	0.202/0.152
Types of interpersonal relationships	Dominating and leading	0.293 */0.035	0.202/0.152
	Collaborative and conventional	0.206/0.144	0.379 **/0.006
	Responsible and generous	0.219/0.119	0.279 */0.045
Emotional attitudes toward oneself, other persons, important phenomena and concepts	Attitude toward civil marriage	0.299 */0.031	0.211/0.134
	Attitude toward registered marriage	−0.124/0.382	0.280 */0.045
	Attitude to own personality	0.438**/0.001	0.026/0.854
	Attitude toward the husband	0.079/0.580	0.576 **/0.000
	Attitude toward the children	0.294 */0.034	−0.034/0.813

The table shows Spearman’s correlation coefficient/p-value; * *p* ≤ 0.05, ** *p* ≤ 0.01.

**Table 2 behavsci-10-00044-t002:** Significant correlation relationships of men’s attitudes to their AP and AP of the spouse with their attitude toward themselves, other persons, and the world.

Dimension	Variables	Men’s Attitude toward Own AP	Attitude of Men to AP of the Spouse
Assessment of components of own AP	Assessment of own face	0.032/0.821	0.304 */0.028
	Assessment of own bodily structure	−0.020/0.891	0.566 **/0.000
	Integral assessment of own AP	0.060/0.672	0.296 */0.033
Expressiveness of social needs	Need to love	0.181/0.199	−0.301 */0.030
	Need to be loved	0.316 */0.023	−0.010/0.943
Emotional attitudes toward oneself, other persons, important phenomena and concepts	Attitude to the family	−0.115/0.415	0.338*/0.014
	Attitude to own personality	0.310 */0.025	0.134/0.345
	Attitude toward the wife	0.181/0.199	0.668 **/0.000
	Attitude toward love	−0.184/0.191	0.503 **/0.000
	Attitude toward the mother	−0.153/0.280	0.275 */0.048

The table shows Spearman’s correlation coefficient/p-value; * *p* ≤ 0.05, ** *p* ≤ 0.01.

**Table 3 behavsci-10-00044-t003:** Results of a comparative analysis partner at spouses being in registered and unregistered marriages.

Variable	Means/Standard Deviations (Group 1)	Means/Standard Deviations (Group 2)	Z Statistics	P-Value	Average Rank of Group 1 (Registered Marriage)	Average Rank of Group 2 (Unregistered Marriage)
**Women**						
Own AP	5.77/2.55	7.19/1.06	−1.938	0.05 *	22.65	30.35
Husband’s AP	5.65/1.83	5.81/1.70	−0.206	0.84	26.08	26.92
**Men**						
Own AP	6.27/2.22	6.77/1.99	−1.231	0.22	24.15	28.85
Wife’s AP	6.50/1.73	5.77/1.53	−2.009	0.05 *	30.62	22.38

The Mann–Whitney criterion, * *p* ≤ 0.05.

**Table 4 behavsci-10-00044-t004:** Results of a comparative analysis of attitude toward own AP in women and attitude toward AP of the wife in men with different durations of the registered marriage.

Variable	Criterion Value	P-Value	Average Rank of Group 1 (Marriage Duration from 0.5 to 10 Years)	Average Rank of Group 2 (Marriage Duration from 11 to 20 Years)	Average Rank of Group 3 (Marriage Duration from 21 to 30 Years)	Means/Standard Deviations (Group 1)	Means/Standard Deviations (Group 2)	Means/Standard Deviations (Group 3)
Women in a registered marriage								
Own AP	9.345	0.009 **	17.00	8.40	7.40	7.06/1.73	4.20/1.79	3.20/2.86
Men in a registered marriage								
Partner’s AP	1.036	0.596	13.66	10.90	15.60	6.50/1.83	6.00/1.87	7.00/1.41

The Kruskal–Wallis H-test, ** *p* ≤ 0.01.

## Data Availability

The datasets used and analyzed during the current study are available from the corresponding author on reasonable request.

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
