# Peer review of "Emotional Attitude to Own Appearance and Appearance of the Spouse: Analysis of Relationships with the Relationship of Spouses to Themselves, Others, and the World"

_behavsci, 2020, doi:10.3390/bs10020044_

Round 1

Reviewer 1 Report

This paper addresses an interesting and timely topic in the context of social psychology. The reviewed paper deals with human marriages and explores the relationship between emotional attitude to own appearance and appearance of the spouse. The authors give a really good introduction into this field and assume three hypotheses. The data set consisting of N=52 married couples is analyzed with descriptive and inferential statistics. Empirical results are presented and discussed.

I really enjoyed reading this paper because I think that the topic of this paper is important for our society. The number of divorced marriages is still increasing so that there is a special need for scientific research in this field. What are relevant factors that influence couple’s satisfaction? What can be done to prevent divorced marriages?

However, the paper in its current form is not ready for publication. My biggest overachieving concern with this study relates to its contribution. What is the added value of this paper? I really didn`t get it. The beginning of this paper is quite good but later I felt lost. In the result section, the authors list plenty of results that seem to come out of the blue. The whole result section seems to be exploratory, whereas three confirmatory hypotheses are assumed in the introduction.

I have some comments/suggestions that I hope will help the authors to further develop this line of work:

(1) Abstract (page 1, line 15): The data set consists of N=52 married couples of which nationalities? Please describe your sample in more detail.

(2) Abstract (page 1, line 15): The authors list four psychological instruments/questionnaires in the abstract. However, later in the paper they cite five questionnaires. The “History of relationships in the couple” (reference [16]) is missing in the abstract. Please list all used instruments in the abstract.

(3) Abstract (page 1, line 15): The authors list their psychological instruments/questionnaires only in keywords. Please use full sentences in the abstract.

(4) Abstract (page 1, line 18): The authors missed to describe their statistical methods in the abstract. Please describe your statistical analyses adequately.

(5) Abstract (page 1, line 24): The authors mention a “complex multi-level determination of the attitude towards appearance”. However, the authors did not conduct a multi-level analysis in the result section. Please delete this phrase unless you report results of a multi-level analyses (also known as meta-regression). In these analyses, researches control for level-1 and level-2 variances when the ICC (intra-class-correlation) is large enough (e.g., in clustered data structures).

(6) Introduction (page 1, line 35): Not “As an interaction context” but rather “As one specific interaction context”.

(7) Introduction (page 2, line 82): Please describe your hypotheses and the current study in a separate chapter with appropriate headings, for example 1.1 Introduction, 1.2 Hypotheses, 1.3 The present study.

(8) Introduction (page 2, line 86): In Hypothesis 2, the authors refer to registered and unregistered marriages as stratification factor/group variable for their analyses. However, what is the rationale for this? Please describe in more detail why you assume a difference between registered and unregistered marriages. Moreover, please provide a clear definition for registered and unregistered marriages. What is the difference between both types of marriages? Are registered and unregistered marriages available in every country?

(9) Introduction (page 2, line 86): In summary, the authors assume three hypotheses. However, the third hypothesis seems to be exploratory in nature (“We also decided to check the pattern found […], page 2, line 86). This is not appropriate. Please provide theoretical background for each hypothesis and cite appropriate literature. What is the scientific rationale to test mentioned hypotheses?

(10) Before “2. Methods” (page 2, line 89): A chapter called “The present study” is completely missing. Please add this chapter and discuss study aims (currently provided in the introduction, page 2, line 65), research questions, and hypotheses. What is the added value of this research?

(11) Before “2. Methods” (page 2, line 89): A chapter called “Procedure” is completely missing. Please add this chapter and describe your data collection. Currently, data of 52 married couples seem to come out of the blue. Were data collected with a pen-and-paper questionnaire? With an online survey software? A chapter called “Procedure” is very important so that readers can evaluate internal and external validity of this study.

(12) Methods (page 2, line 92 and 93): Please provide for descriptive statistics mean, standard deviation, minimum and maximum values in the whole paper. Here, please provide mean and standard deviation for duration of marriage and age.

(13) Methods (page 3, line 113): A chapter called “Statistics” is completely missing. Please add this chapter and describe the statistical analyses used in this paper. Currently, the authors describe statistical methods with one sentence only (“For data processing, the following […]”, page 3, line 113). Obviously, this is not appropriate. Please mention which statistical software you used (e.g., SPSS, R, Mplus, Stata, SAS) and which version of it. Please provide more information about types of variables/scales (dichotomous, categorical or metric?). This is important for interpreting correlations in the result section. Please mention your significance level. I guessed the typical 5%, correct?

(14) Results (page 3, line 121 and 122): Please provide information about used statistics and p-values in footnotes and not in table headings.

(15) Results (page 3, below line 122): Obviously, Table 1 as well as Table 2 represent correlation matrices. However, I was confused about empty cells. Why not providing full correlation matrices? The authors might argue that not all correlations are important. However, for transparency reasons, I urge the authors to provide full correlation matrices for Table 1 as well as for Table 2. To guide readers’ attention, important correlation can be marked in bold.

(16) Results (page 3, below line 122): I was confused about labels in Table 1 and Table 2 (“Studied psychological variables” and “Analysis parameters”). Please use “Dimension” instead of “Studied psychological variables” and “Variables” instead of “Analysis parameters”.

(17) Results (page 4, line 126): Sometimes “tables 1 and 2” (page 4, line 126), sometimes “Tables 1-2” (page 3, line 119). Please keep a consistent writing style and use capital letters for Tables.

(18) Results (page 4, line 126): I really appreciate that authors have structured their result section along the mentioned three hypotheses (chapter 3.1, 3.2, and 3.3). However, I felt really lost when I read the description of empirical results. The authors list plenty of results that seem to come out of the blue. The whole result section seems to be exploratory, whereas three confirmatory hypotheses are assumed in the introduction. Please shorten this section and guide the reader through your empirical findings. Report only results that are related to your hypotheses. Table 1 and Table 2 contain all calculated correlations but, please, discuss only such results in the text that are really relevant for answering your research questions and your hypotheses. Please add after each statement the relevant correlation and the p-value like you already did on page 6, line 200 (i.e., provide scientific evidence for your statement). For example: “[…] the emotional attitude towards the partner’s AP is closely connected with the attitude towards the partner (r=XX, p=xx).” (page 4, line 129).

(19) Results (page 5, line 173): As before (Table 1 and 2) please provide notes about statistics and p-values in footnotes below Table 3. The same is true for Table 4 (page 5, line 186 and 187).

(20) Results (page 5, 173): The authors analyzed characteristics between men and women with a non-parametric Mann-Whitney-U test. However, they missed to present means and standard deviations for outcome variables (e.g., attitude to own AP). Please add means and standard deviations in Table 3. The same is true for Table 4 (page 6, above line 188).

(21) Results (page 5, below line 173): I was confused about labels in Table 3 (“Parameters” and “Significance level”). Please use “Variable” instead of “Parameters” and “p-value” instead of “Significance level”.

(22) Results (page 6, line 200): Sometimes “registered and unregistered marriages” (page 5, line 162), sometimes “official and unofficial marriage” (page 6, line 200). Please keep a consistent writing style and use one terminology only. Otherwise, the reader is confused about different wordings for the same thing.

(23) Discussion (page 7, line 237): A chapter called “Limitations” is completely missing. Please add this chapter in the discussion and describe the limitations of presented study. Please mention your study design and data collection.

(24) Discussion (page 7, line 237): A chapter called “Practical Implications” is completely missing. Please add this chapter in the discussion and discuss practical implication of your study. Which practical benefits can be gained from the empirical findings? What is the added value of your research? Please describe in detail and provide an outlook on future research in this field.

Author Response

The authors thank the reviewer for a detailed and comprehensive analysis of the paper, and tried to take into account all the comments when finalizing the article.

(1) Abstract (page 1, line 15): The data set consists of N=52 married couples of which nationalities? Please describe your sample in more detail.

The sample in the annotation is described in more detail (nationality, citizenship, age).

(2) Abstract (page 1, line 15): The authors list four psychological instruments/questionnaires in the abstract. However, later in the paper they cite five questionnaires. The “History of relationships in the couple” (reference [16]) is missing in the abstract. Please list all used instruments in the abstract.

All research methods were inserted in the annotation

(3) Abstract (page 1, line 15): The authors list their psychological instruments/questionnaires only in keywords. Please use full sentences in the abstract.

A full description of the methods were inserted in the annotation

(4) Abstract (page 1, line 18): The authors missed to describe their statistical methods in the abstract. Please describe your statistical analyses adequately.

Statistical methods were inserted in the annotation

(5) Abstract (page 1, line 24): The authors mention a “complex multi-level determination of the attitude towards appearance”. However, the authors did not conduct a multi-level analysis in the result section. Please delete this phrase unless you report results of a multi-level analyses (also known as meta-regression). In these analyses, researches control for level-1 and level-2 variances when the ICC (intra-class-correlation) is large enough (e.g., in clustered data structures).

The phrase «The conducted study shows the complex multi-level determination of the attitude towards appearance»  has been removed from the annotation

(6) Introduction (page 1, line 35): Not “As an interaction context” but rather “As one specific interaction context”.

The phrase” As an interaction context "has been replaced with “As one specific interaction context".

(7) Introduction (page 2, line 82): Please describe your hypotheses and the current study in a separate chapter with appropriate headings, for example 1.1 Introduction, 1.2 Hypotheses, 1.3 The present study.

The structure of the article has been changed; were added a section Hypotheses

(8) Introduction (page 2, line 86): In Hypothesis 2, the authors refer to registered and unregistered marriages as stratification factor/group variable for their analyses. However, what is the rationale for this? Please describe in more detail why you assume a difference between registered and unregistered marriages. Moreover, please provide a clear definition for registered and unregistered marriages. What is the difference between both types of marriages? Are registered and unregistered marriages available in every country?

The definition of unregistered marriag is given, the statistics of unregistered marriages in Russia are given, and it is stated that the increase in unregistered marriages is a trend for developed countries.

(9) Introduction (page 2, line 86): In summary, the authors assume three hypotheses. However, the third hypothesis seems to be exploratory in nature (“We also decided to check the pattern found […], page 2, line 86). This is not appropriate. Please provide theoretical background for each hypothesis and cite appropriate literature. What is the scientific rationale to test mentioned hypotheses?

The third hypothesis is reformulated. The theoretical basis, references to the relevant literature, and the rationale for each hypothesis are presented in the introduction and paragraph 2. Present study. The justification for the second and third hypotheses has been added.

(10) Before “2. Methods” (page 2, line 89): A chapter called “The present study” is completely missing. Please add this chapter and discuss study aims (currently provided in the introduction, page 2, line 65), research questions, and hypotheses. What is the added value of this research?

In the paper was inserted the section " The present study”, which includes the goals and objectives of the study

(11) Before “2. Methods” (page 2, line 89): A chapter called “Procedure” is completely missing. Please add this chapter and describe your data collection. Currently, data of 52 married couples seem to come out of the blue. Were data collected with a pen-and-paper questionnaire? With an online survey software? A chapter called “Procedure” is very important so that readers can evaluate internal and external validity of this study.

The “Procedure” item was added to the article, which takes into account the reviewer's comments.

 (12) Methods (page 2, line 92 and 93): Please provide for descriptive statistics mean, standard deviation, minimum and maximum values in the whole paper. Here, please provide mean and standard deviation for duration of marriage and age.

The sample is described in more detail with an indication of the mean and standard deviations for the duration of marriage and age

(13) Methods (page 3, line 113): A chapter called “Statistics” is completely missing. Please add this chapter and describe the statistical analyses used in this paper. Currently, the authors describe statistical methods with one sentence only (“For data processing, the following […]”, page 3, line 113). Obviously, this is not appropriate. Please mention which statistical software you used (e.g., SPSS, R, Mplus, Stata, SAS) and which version of it. Please provide more information about types of variables/scales (dichotomous, categorical or metric?). This is important for interpreting correlations in the result section. Please mention your significance level. I guessed the typical 5%, correct?

The “Statistics” item was added to the article, which takes into account the reviewer's comments.

 (14) Results (page 3, line 121 and 122): Please provide information about used statistics and p-values in footnotes and not in table headings.

Information about the method and the significance level is given in the footnotes to the table

(15) Results (page 3, below line 122): Obviously, Table 1 as well as Table 2 represent correlation matrices. However, I was confused about empty cells. Why not providing full correlation matrices? The authors might argue that not all correlations are important. However, for transparency reasons, I urge the authors to provide full correlation matrices for Table 1 as well as for Table 2. To guide readers’ attention, important correlation can be marked in bold.

The lines left empty contain missing, non-significant correlation coefficients. However, it is not possible to provide a full correlation matrix in this variant of paper (that takes into account all the correlations obtained), since in this case the volume of the article would be  increased by several pages. According to the terms of publication in the special collection ECP2019 a paper must not exceed 8 pages. Significant correlations in Tables 1 and 2 are marked with * and **

(16) Results (page 3, below line 122): I was confused about labels in Table 1 and Table 2 (“Studied psychological variables” and “Analysis parameters”). Please use “Dimension” instead of “Studied psychological variables” and “Variables” instead of “Analysis parameters”.

Сhanges has been made

(17) Results (page 4, line 126): Sometimes “tables 1 and 2” (page 4, line 126), sometimes “Tables 1-2” (page 3, line 119). Please keep a consistent writing style and use capital letters for Tables.

Changes has been made

(18) Results (page 4, line 126): I really appreciate that authors have structured their result section along the mentioned three hypotheses (chapter 3.1, 3.2, and 3.3). However, I felt really lost when I read the description of empirical results. The authors list plenty of results that seem to come out of the blue. The whole result section seems to be exploratory, whereas three confirmatory hypotheses are assumed in the introduction. Please shorten this section and guide the reader through your empirical findings. Report only results that are related to your hypotheses. Table 1 and Table 2 contain all calculated correlations but, please, discuss only such results in the text that are really relevant for answering your research questions and your hypotheses. Please add after each statement the relevant correlation and the p-value like you already did on page 6, line 200 (i.e., provide scientific evidence for your statement). For example: “[…] the emotional attitude towards the partner’s AP is closely connected with the attitude towards the partner (r=XX, p=xx).” (page 4, line 129).

The section has been reworked in accordance with the reviewer's comments: it has been shortened, and an argument has been added after each statement.

(19) Results (page 5, line 173): As before (Table 1 and 2) please provide notes about statistics and p-values in footnotes below Table 3. The same is true for Table 4 (page 5, line 186 and 187).

Information about the method and the significance level is given in the footnotes to the table

(20) Results (page 5, 173): The authors analyzed characteristics between men and women with a non-parametric Mann-Whitney-U test. However, they missed to present means and standard deviations for outcome variables (e.g., attitude to own AP). Please add means and standard deviations in Table 3. The same is true for Table 4 (page 6, above line 188).

Means and standard deviations has been included in tables 3 and 4

(21) Results (page 5, below line 173): I was confused about labels in Table 3 (“Parameters” and “Significance level”). Please use “Variable” instead of “Parameters” and “p-value” instead of “Significance level”.

Correction has been made

(22) Results (page 6, line 200): Sometimes “registered and unregistered marriages” (page 5, line 162), sometimes “official and unofficial marriage” (page 6, line 200). Please keep a consistent writing style and use one terminology only. Otherwise, the reader is confused about different wordings for the same thing.

Terminology “registered and unregistered marriages” are left everywhere in the text

(23) Discussion (page 7, line 237): A chapter called “Limitations” is completely missing. Please add this chapter in the discussion and describe the limitations of presented study. Please mention your study design and data collection.

The “Limitations” section is inserted at the end of the Discussion Chapter

(24) Discussion (page 7, line 237): A chapter called “Practical Implications” is completely missing. Please add this chapter in the discussion and discuss practical implication of your study. Which practical benefits can be gained from the empirical findings? What is the added value of your research? Please describe in detail and provide an outlook on future research in this field.

The «Practical Implications» section is inserted at the end of the Discussion Chapter

Reviewer 2 Report

In this research manuscript authors investigated the appearance attitude towards the own personality and other persons in men and women being or not married.

The research plan is clearly described and performed. English language is fine.

Author Response

The authors thank the reviewer for a detailed and comprehensive analysis of the paper

Reviewer 3 Report

The article meets the requirements of originality. The authors proposed a new approach to solving the scientific problem.

Author Response

(The authors gave the same response as above.)

Round 2

Reviewer 1 Report

I thank the authors for submitting a revised version of their manuscript “Emotional Attitude to Own Appearance and Appearance of the Spouse: Analysis of Relationships with the Relationship of Spouses to Themselves, Others, the World”. After intensive reading, I confirm that the quality of the manuscript increased a lot! I really appreciate the efforts of the authors and their ideas in the current version. However, there are still (minor) aspects in the manuscript that should be corrected before final publication.

I have some comments/suggestions that I hope will help the authors to further develop this line of work:

(1) Abstract (page 1, line 17): The authors now include all empirical instruments in the abstract, thanks a lot! However, please do not mention the names of the authors in the abstract. Please delete all brackets with the names (e.g., T.A.Shkurko, M. A. Lomova). Note, however, mentioning the authors names in the full-text is correct, but not in the abstract.

(2) Abstract (page 1, line 20): The authors now mention the statistical analyses used in this paper, thanks a lot! However, please write full sentences and not keywords. Example: not “Spearman correlation analysis, […]” but rather “Empirical data were analyzed with Spearman correlation analysis, […]”.

(3) 2.5 Statistics (page 3, line 135): The sentence “Data analysis was performed at significance levels of 0.05 and 0.01” is not appropriate because you can only refer to one significance level at a time. Probably you use the conventional α=0.05 significance level. Hence, every p-value equal or below this level will be declared as statistically significant. So please write instead “p-values ≤0.05 are considered as statistically significant”.

(4) Table 1 (page 4, line 144): “The Table shows Spearman’s correlation coefficient/coefficient significance level” is statistically wrong. Reason: The significance level (here: 5%) is something different as the p-value. I guess, the authors refer to the p-value of each correlation coefficient. If this is correct, please write “The Table shows Spearman’s correlation coefficient/p-value” instead. Please correct this in the whole manuscript.

(5) Applies for the whole manuscript: Since this is an English text, please use the dot and not the comma as a decimal separator (e.g., not p=0,001 but rather p=0.001) Please correct this in the whole manuscript (full-text, Tables, and Figures). The comma is normally used to indicate numbers in the thousands-level (e.g. 15,000 equals fifteen thousand).

(6) Limitations (page 7, line 274): The authors now include a limitation section, thanks a lot. However, this section is very short and the beginning written in keywords. Please discuss in more detail why a small sample size, a fairly large range of respondents’ age, and duration of cohabitation is a limitation of the presented results. In the next sentence (page 7, line 275) the authors go in the right direction (i.e., limited external validity of their empirical findings). Please enlarge the limitation section and think of other aspects like data collection (randomized sample? If not, what is the consequence of that? Does Spearman correlation coefficients imply causality? If not, please discuss this and other aspects of your study in more detail).

(7) Practical Implications (page 7, line 278): The authors now include a section with practical implications, thanks a lot. The authors write “Further research suggests expanding the sample of respondents and focusing on spouses of different ages” (page 7, line 281). This makes no sense compared to the presented limitations (see page 7, line 274, “[…] a fairly large range of respondents’ age […]”. On the one hand, a fairly large range of respondents’ age is a limitation of this study, on the other hand the authors want to focus on spouses of different ages? Please clarify or revise this statement.

Author Response

I thank the authors for submitting a revised version of their manuscript “Emotional Attitude to Own Appearance and Appearance of the Spouse: Analysis of Relationships with the Relationship of Spouses to Themselves, Others, the World”. After intensive reading, I confirm that the quality of the manuscript increased a lot! I really appreciate the efforts of the authors and their ideas in the current version. However, there are still (minor) aspects in the manuscript that should be corrected before final publication.

The authors express their great gratitude to the reviewer for valuable advice and comments that have significantly improved the text of the manuscript.

I have some comments/suggestions that I hope will help the authors to further develop this line of work:

 (1) Abstract (page 1, line 17): The authors now include all empirical instruments in the abstract, thanks a lot! However, please do not mention the names of the authors in the abstract. Please delete all brackets with the names (e.g., T.A.Shkurko, M. A. Lomova). Note, however, mentioning the authors names in the full-text is correct, but not in the abstract.

Mentioning of authors ' names is excluded from the abstract.

(2) Abstract (page 1, line 20): The authors now mention the statistical analyses used in this paper, thanks a lot! However, please write full sentences and not keywords. Example: not “Spearman correlation analysis, […]” but rather “Empirical data were analyzed with Spearman correlation analysis, […]”.

The phrase «Empirical data were analyzed with Spearman correlation analysis, Mann-Whitney U-test, Kruskal-Wallis H-test» was inserted in the annotation.

 (3) 2.5 Statistics (page 3, line 135): The sentence “Data analysis was performed at significance levels of 0.05 and 0.01” is not appropriate because you can only refer to one significance level at a time. Probably you use the conventional α=0.05 significance level. Hence, every p-value equal or below this level will be declared as statistically significant. So please write instead “p-values ≤0.05 are considered as statistically significant”.

The authors thank the reviewer for explanations and recommendations! The corresponding proposal was included in the text.

(4) Table 1 (page 4, line 144): “The Table shows Spearman’s correlation coefficient/coefficient significance level” is statistically wrong. Reason: The significance level (here: 5%) is something different as the p-value. I guess, the authors refer to the p-value of each correlation coefficient. If this is correct, please write “The Table shows Spearman’s correlation coefficient/p-value” instead. Please correct this in the whole manuscript.

The authors thank the reviewer for the clarification, the phrase «coefficient significance level» has been corrected to « p-value»

(5) Applies for the whole manuscript: Since this is an English text, please use the dot and not the comma as a decimal separator (e.g., not p=0,001 but rather p=0.001) Please correct this in the whole manuscript (full-text, Tables, and Figures). The comma is normally used to indicate numbers in the thousands-level (e.g. 15,000 equals fifteen thousand).

Thanks for the recommendation! The entire test now uses a comma as the decimal separator.

(6) Limitations (page 7, line 274): The authors now include a limitation section, thanks a lot. However, this section is very short and the beginning written in keywords. Please discuss in more detail why a small sample size, a fairly large range of respondents’ age, and duration of cohabitation is a limitation of the presented results. In the next sentence (page 7, line 275) the authors go in the right direction (i.e., limited external validity of their empirical findings). Please enlarge the limitation section and think of other aspects like data collection (randomized sample? If not, what is the consequence of that? Does Spearman correlation coefficients imply causality? If not, please discuss this and other aspects of your study in more detail).

The authors completely rewritten the section “Limitations” in accordance with the reviewer's recommendations

(7) Practical Implications (page 7, line 278): The authors now include a section with practical implications, thanks a lot. The authors write “Further research suggests expanding the sample of respondents and focusing on spouses of different ages” (page 7, line 281). This makes no sense compared to the presented limitations (see page 7, line 274, “[…] a fairly large range of respondents’ age […]”. On the one hand, a fairly large range of respondents’ age is a limitation of this study, on the other hand the authors want to focus on spouses of different ages? Please clarify or revise this statement.

The last sentence was excluded from the section "Practical Implications", since its clarification is already given above, in the section “Limitations”